# Coseismic Deformation and Fault Inversion of the 2017 Jiuzhaigou Ms 7.0 Earthquake: Constraints from Steerable Pyramid and InSAR Observations

**Wenshu Peng \*, Xuri Huang and Zegen Wang**

School of Earth Science and Technology, Southwest Petroleum University, Chengdu 610500, China
* Correspondence: 202111000044@stu.swpu.edu.cn; Tel.: +86-18811589039

**Abstract:** The 8 August 2017 Ms 7.0 Jiuzhaigou earthquake was generated in the transition zone between the Tazang fault, Huya fault, and Minjiang fault, all being part of the East Kunlun fault system. In this study, two pairs of SAR (synthetic aperture radar) data from Sentinel-1 satellite were used to derive the surface displacement observations along the satellite line-of-sight (LOS) directions using the differential interferometric SAR (D-InSAR) method. A steerable pyramid filtering method (i.e., a method for a linear multiscale, multidirectional decomposition and filtering technology) was proposed to optimize and enhance the geological features from interferometric image and coseismic deformation field. The 3D deformation was derived under the constraint of the combined D-InSAR and MAI method. The small baseline subset InSAR (SBAS-InSAR) time series method was used to obtain the cumulative deformation across the fault system. Fault slip inversion results from interferogram of InSAR indicate that the 2017 Jiuzhaigou earthquake was dominated by left-lateral slip, the surface movement was dominated by horizontal deformation, the vertical deformation was small, and the coseismic deformation variable in the east–west direction was the largest, with a maximum deformation of 0.2 m to the east and 0.14 m to the west. The maximum slip is about 77 cm, which is located at a depth of 9 km. The moment magnitude obtained by inversion is Mw 6.6, and the seismic fault is the Huya fault.

**Keywords:** D-InSAR; SBAS-InSAR; steerable pyramid filtering; interferogram; coseismic deformation field; Xianshuihe fault; Jiuzhaigou Ms 7.0 earthquake

## 1. Introduction

On 8 August 2017, an Ms 7.0 earthquake that occurred in Jiuzhaigou county, Sichuan Province (33.2°N, 103.82°E), was one of the most destructive and intense earthquakes in China in recent years. The earthquake was another strong earthquake that occurred at the boundary of the Bayan Har block since the 2013 Lushan earthquake, causing huge casualties and property damage. This earthquake was generated in the junction between the Tazang fault (TZF), Minjiang fault (MJF), and the Huya fault (HF), both being part of the eastern section of the East Kunlun fault system. This fault system is generally characterized by left-lateral slip and is the northeastern boundary of the Bayan Har block.

From the perspective of regional tectonics, due to the east–south movement of the Bayan Har block being blocked by the South China block, the area is undergoing strong deformation. The eastern end of the East Kunlun fault experiences a large angle of deflection to the south in the strike direction, and forms a broom-like spread of multiple branch faults, which intersect with the Longriba fault, the Minjiang fault, and the Huya fault, and the nature of the fault movement gradually changes from strike-slip to reverse thrust [1]. That is, the Tazang fault is a Holocene active fault, and its western section is dominated by strike-slip movement, and the eastern section is dominated by backlash: the Longri Dam fault is a right-handed strike-slip and backlash nature, which has been an active fault since the late Pleistocene; the Minjiang fault is a Holocene active fault with left-handed strike-slip

motion dominated by backlash; the Huya fault gradually changed from left-handed slide to backlash mode from north to south, and many strong earthquakes have occurred in historical records.

InSAR technology is an emerging new remote sensing technology developed in the past two decades. It originated from the "Young's Double Slit Interference Experiment" conducted by Thomas Young in 1801. D-InSAR technology, that is, differential interferometric synthetic aperture radar technology, is developed based on the traditional InSAR [2]. It uses surface SAR image data obtained at different times in the same area to perform differential interference processing to remove the common quantities in the two observation phases (flat ground effect, terrain phase, atmospheric delay, etc.) and to obtain the deformation phase for surface deformation detection. With its high accuracy and reliability of the surface deformation, the D-InSAR method has been widely applied to earthquake coseismic deformation analysis. Many scholars have carried out research on earthquakes through the deformation results of D-InSAR. Grabriel et al. [3] used D-InSAR technology to monitor large-scale deformation regions and obtained centimeter-level accuracy results. Massonnet et al. [4] obtained the deformation field of the 1992 Landers earthquake (M = 7.2) using ERS1SAR data, which is consistent with the results of traditional monitoring data and elastic deformation model.

However, due to the interference of noise, the fringes in the D-InSAR interferogram images may not be obvious, or may even be completely submerged in the noise. Therefore, the filtering of interference fringe patterns is the key research goal of scientists around the world. In 1998, Lee et al. [5] proposed a phase filtering method to achieve adaptive intensity filtering of the InSAR phase map in the complex plane. The large-scale filtering in the low-coherence region effectively filters out the noise while avoiding the high-coherence region. In the same year, Goldstein et al. [2] proposed an adaptive filter method based on the phase interferogram spectrum, which transforms a small block of complex interferograms into the spectral domain and uses an amplitude smoothing filter in the frequency domain to adjust the phase spectrum. In the next two years, some attempts were made to improve the effect of the Goldstein filter and enhance the local adaptability of the filter [3–5]. In 2006, Wu et al. [6] proposed a method using the local fringe frequency of the InSAR interferometric phase image to determine its direction to perform targeted adaptive filtering, which greatly improved the quality of phase image fringe edge detail preservation. In 2013, Chao et al. [7] made further optimizations for the Lee adaptive filter to improve the performance of the original filter. In the same year, Fu et al. [8] improved the size and direction of the filtering window to make corresponding adjustments to achieve the purpose of adaptive interferometric phase pattern filtering. In 2015, Song et al. [9] developed an improved version of the original classic Goldstein filter with a certain degree of optimization by using the method of adapting the pixel values of surrounding blocks. In the same year, Wang et al. [10] proposed a dual-domain phase filter based on an improved Baran filter and an adaptive mean filter.

Based on the above problems, this paper proposes the steerable pyramid filtering (SPF) method for InSAR interferogram images processing. SPF is a linear multiscale, multidirectional decomposition and filtering technology [11]. In this method, the filter can be applied in different directions and different scales. This provides a niche way to optimize and enhance the geological features from data such as coseismic deformation field and DEM. The results show that the SPF method has significant denoising and edge-enhancement effects on the interferogram fringe and deformation field. In order to solve the problem that traditional D-InSAR technology cannot obtain the temporal evolution of surface deformation due to the influence of atmospheric disturbance, space–time correlation, and other factors, we applied SBAS technology to obtain the persistent deformation rate of the study area for a long time after the earthquake [12]. The possible causes of the Jiuzhaigou earthquake based on the evidence of structural geology are discussed. The results provide evidence to support the accuracy of the SPF method in differential interference image processing results.

## 2. Study Area and Data

### 2.1. Study Area

The epicenter of the Jiuzhaigou earthquake is located at the intersection area of the Tazang fault (TZF), the Minjiang fault (MJF), and the Huya fault (HYF) (Figure 1). With main seismogenic structure shown in Figure 2, this area is a historical earthquake zone [13]. Among them, the HYF gradually changed from a left-lateral strike-slip to a thrust movement from north to south. Historically, in 1973, the Songpan East M6.5 earthquake occurred in the southern part of the northern section of the TZF [14]; while the late Quaternary activity of the TZF was segmented and multiperiodic, the post-earthquake slide of the fault was mainly left stable slip, with a rate of 10.27 mm/a, strike slip was the main movement, and the eastern section was dominated by thrusting [15]. In the southern section of the HYF, two earthquakes with magnitude 7.2 and one earthquake with magnitude 6.7 occurred in 1976; after the earthquakes, the relative deformation of the two plates of HYF was 302 mm, the deformation gradient around the fault is 0.6 mm/km per year, and the relative motion rate is an average of 20 mm per year [16].

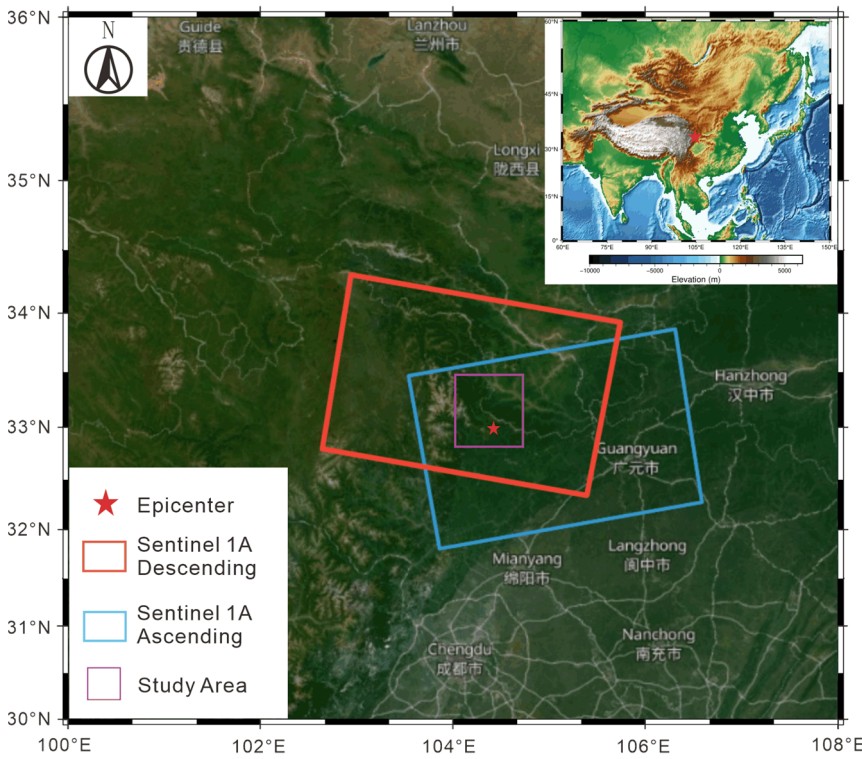

**Figure 1.** The location of the study area and coverage areas of SAR datasets.

### 2.2. Data and Processing

Table 1 presents the basic information of the SAR data used in the D-InSAR processing; the two-track method is used to generate a coseismic deformation interferogram. The terrain phase cancellation uses the 30 m resolution SRTM digital elevation model published by NASA. In order to suppress the noise, 25 × 6 (slant range and azimuth) multilooking is performed on the interferogram in the InSAR data processing. The interferogram is filtered by the weighted power spectrum method twice. The filter window is set to 128 × 128 and 32 × 32, respectively. This filter window setting can greatly improve the coherence of the interferogram [17]. The phase unwrapping uses the least-cost flow algorithm. Removal of residual orbital phases from interferograms is achieved by quadratic polynomial fitting. For the phase delay caused by the vertical stratification of atmospheric water vapor, an atmospheric phase delay model is established based on the existing digital elevation model. The phase delay is removed from the original interferogram.

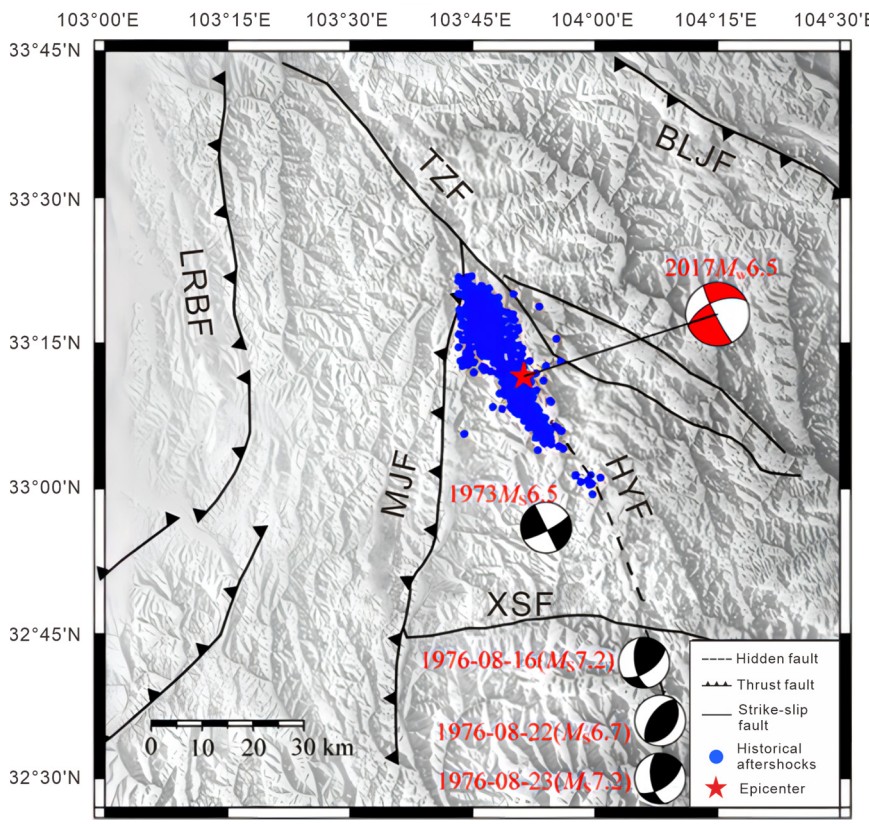

**Figure 2.** Map with topography, faults, and major historical earthquakes of the study area.

**Table 1.** Basic information of the SAR data used in the D-InSAR interferogram.

| Sensor | Orbit Direction | Master–Slave Date | Track | Spatial Perpendicular Baseline (m) | Temporal Baseline (day) | Incident Angle (○) |
|---|---|---|---|---|---|---|
| Sentinel-1A | Ascending | 20170730–20170811 | T128 | 36.0 | 12 | 39.2 |
| Sentinel-1A | Descending | 20170806–20170818 | T62 | 34.0 | 12 | 39.2 |

SBAS-InSAR is an InSAR time series method based on multisubject images. It combines large amounts of SAR data into interference subsets with multiple master images through the principle of short baseline. The baseline length of the interference pair within each subset is lower than the critical baseline value, the time baseline is as short as possible, and the SAR image baseline distance between sets is large. In this way, the incoherence in time and space is overcome. In this study, the SBAS-InSAR technique was used to observe the possible cumulative displacement in the Jiuzhaigou earthquake area in half a year after the earthquake.

Sixteen scenes of images from the ascending orbit of Sentinel-1A were obtained (Table 2), performing multiview, filtering, phase unwrapping, and interferometric pair selection on the images, and finally obtaining 25 interferometric pairs. In order to improve the processing accuracy of Sentinel-1A image data, this study uses satellite precise orbit data. The DEM data used in the experiment is the ALOS digital surface model "ALOS World3D-30m" provided by the JAXA Corporation of Japan. The temporal and spatial baseline relationship of the interference image pair is shown in Figure 3.

**Table 2.** Basic information of the SAR data used in the SBAS-InSAR interferogram.

| Sensor | Imaging Date | Orbit Direction | Imaging Mode | Polarization Mode | Cumulative Time Baseline (Day) |
|---|---|---|---|---|---|
| Sentinel-1A | 20170818 | Descending | IW | VV | 0 |
| Sentinel-1A | 20170830 | Descending | IW | VV | 12 |
| Sentinel-1A | 20170911 | Descending | IW | VV | 24 |
| Sentinel-1A | 20170923 | Descending | IW | VV | 36 |
| Sentinel-1A | 20171005 | Descending | IW | VV | 48 |
| Sentinel-1A | 20171017 | Descending | IW | VV | 60 |
| Sentinel-1A | 20171110 | Descending | IW | VV | 84 |
| Sentinel-1A | 20171122 | Descending | IW | VV | 96 |
| Sentinel-1A | 20171204 | Descending | IW | VV | 108 |
| Sentinel-1A | 20171216 | Descending | IW | VV | 120 |
| Sentinel-1A | 20171228 | Descending | IW | VV | 132 |
| Sentinel-1A | 20180109 | Descending | IW | VV | 144 |
| Sentinel-1A | 20180121 | Descending | IW | VV | 156 |
| Sentinel-1A | 20180202 | Descending | IW | VV | 168 |
| Sentinel-1A | 20180214 | Descending | IW | VV | 180 |
| Sentinel-1A | 20180226 | Descending | IW | VV | 192 |

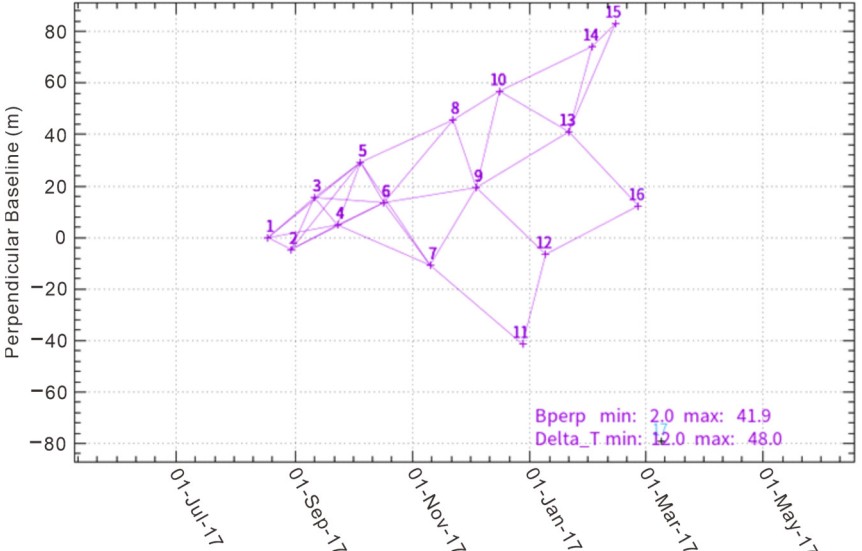

**Figure 3.** Interferometric image space–time baseline map.

## 3. Methods

### 3.1. Steerable Pyramid Filtering Method

Steerable pyramid is an effective but conceptually simple structure to explain images at multiple resolutions, and it is frequently used for image segmentation [18]. The pyramid means a series of images arranged in a pyramid structure derived from the same original image with decreasing resolution. It is obtained by echelon downsampling, and continues this sampling until a certain termination condition is reached. When performing directional steerable filtering on an image, a convolution operation is applied, that is, the input image is convolved with several base filters in different directions to obtain corresponding base-filtered images. Then, the corresponding weights are assigned to the base-direction-filtered images [19]. The coefficients of weights and base-direction-filtered images are multiplied and added to obtain the final filtered image. Two main functions of the steerable pyramid filter are multiscale decomposition and reconstruction, and directionally steerable filtering.

#### 3.1.1. Multiscale Decomposition and Reconstruction

As shown in Figure 4, the image pyramid decomposes the image into a series of bottom-up images with gradual changes in scale or resolution through the structural relationship between large-scale and small-scale signals. From top to bottom, the scale

of the pyramid composition map changes from small to large. Small scales can reflect the detailed features of the image, while large scales can better reflect the macroscopic characteristics of the image. The key to build an image pyramid is to sample or interpolate the rows and columns of the image data according to a certain ratio. Appropriate scale can not only ensure the accuracy of multiscale construction, but also reduce the computational complexity of image data.

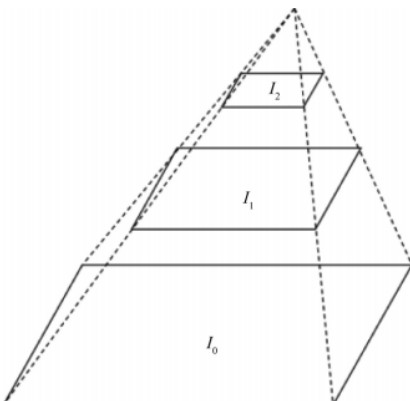

**Figure 4.** Image pyramid conceptual structure diagram.

3.1.2. Directionally Steerable Filter

First proposed by Freeman and Adelson, the directionally steerable filter is defined as a special filter in any direction that can be linearly combined by a set of base filters [20]. It has the function of arbitrary rotation, is an extension of the directional filter, is integrable and derivable in direction, and has continuity [21]. More importantly, it avoids the errors caused by the use of discrete interpolation (which are introduced during the rotation of the original directional filter), so that high accuracy can be obtained. The filter is steerable, and the processed result has a low calculation amount and high filtering accuracy.

The function expression of the controllable filter is as follows:

$$f^\theta(x,y) = \sum_{j=1}^{N} [k_j(\theta)\, f^{\theta j}(x,y)] \tag{1}$$

$f^\theta(x,y)$—function of the controllable filter in the $\theta$ direction;
$k_j(\theta)$—interpolation function in the $\theta$ direction;
$f^{\theta j}(x,y)$—basis function in the $\theta$ direction;
$\theta$—angle of rotation;
$j$—the number of base filters.

In order to reduce the computational amount of the base filter, and to solve the problems of $f^{\theta j}(x,y)$ and interpolation function $k_j(\theta)$ selection, the Cartesian coordinate system is converted to a polar coordinate system, and $f^\theta(x,y)$ is expressed in polar coordinates as follows:

$$f^\theta(r,\varphi) = \sum_{j=1}^{N} [k_j(\theta)\, g_j(r,\varphi)] \tag{2}$$

$$r = \sqrt{x^2 + y^2} \tag{3}$$

$$\varphi = arg(x,y) \tag{4}$$

where $g_j(r,\varphi)$ is the basis function in the direction $\theta$ at polar coordinates.

The structure and processing of direction-controllable filters are shown in Figure 5. First, input the image, convolve the input image with a set of base filters (three different

directions), then multiply the directional filter image by the corresponding interpolation function, and then add the results to obtain the final filtered image.

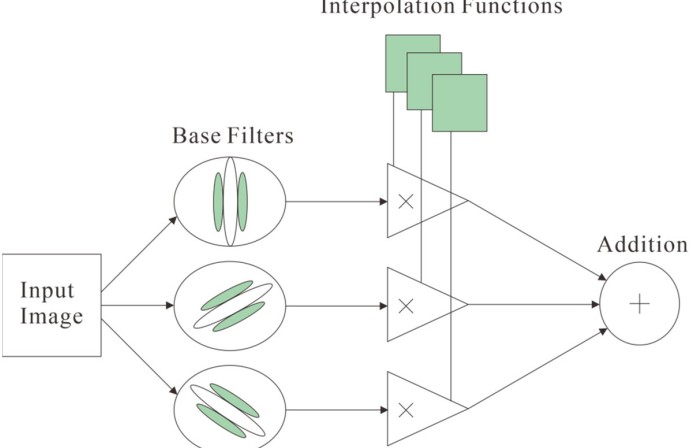

**Figure 5.** Schematic diagram of the structure and processing process of the direction-controllable filter.

In this study, the SPF method was processed on the interferogram image and deformation field to reduce noise and optimize and enhance the boundary of interference pattern and deformation field. The result was validated by comparing with the results of previous works of other scholars.

*3.2. Decomposition of Three-Dimensional Deformation Field*

In the figure, $d_V$, $d_N$, and $d_E$ are the deformations in the vertical, north–south, and east–west directions, respectively, and specify that upward, due north, and due east are positive; $d_{los}$ and $d_{az}$ are, respectively, the deformation of the line-of-sight and the azimuth directions. The observation equations for the three-dimensional deformation of earthquakes can be obtained along the satellite orbit as positive, and the unified positive direction can obtain the three-dimensional deformation of earthquakes.

Assuming that the results of satellite ascending and descending interferometric measurements are $d_{LOSasc}$ and $d_{LOSdes}$, and the simulated north–south component of the deformation is $d_N$, Equation (5) can be further expressed as:

$$\begin{cases} d_{losA} = d_V cos\theta_A + d_N sin\theta_A sin\alpha_A + d_E cos\alpha_A sin\theta_A \\ d_{losD} = d_V cos\theta_D + d_N sin\theta_D sin\alpha_D + d_E cos\alpha_D sin\theta_D \\ d_{azA} = d_N cos\alpha_A + +d_E sin\alpha_A \\ d_{azD} = d_N cos\alpha_D + +d_E sin\alpha_D \end{cases} \quad (5)$$

where $\theta_A$ is the angle of incidence of the ascending orbit, $\theta_D$ is the angle of incidence of the descending orbit, $\alpha_A$ is the azimuth angle of the satellite in the ascending orbit, and $\alpha_D$ is the azimuth angle of the satellite in the descending orbit. Observation vector $D$:

$$D = (d_{losA}, d_{losD}, d_{azA}, d_{azD}) \quad (6)$$

Obtain the error equation:

$$V = CX - D \quad (7)$$

where $C$ is the coefficient matrix of the observation equation, and vector $V$ is the error of each observation, i.e.,

$$V = (V_{losA}, V_{losD}, V_{azA}, V_{azD})^T \quad (8)$$

The three-dimensional displacement field can be calculated by the weighted least-squares principle:

$$X = \left(A^T P A\right)^{-1} A^T P D \quad (9)$$

*P* is the observation weight matrix, because the DInSAR measurement error and the azimuth deformation error of MAI are at the centimeter level, and the accuracy of the four observations is approximately considered to be equal, that is, the weight matrix *P* is the scalar matrix.

## 4. Results

### 4.1. D-InSAR Interferogram Image and Displacement Field

The descending orbit interferogram (Figure 6b) shows a completely opposite deformation situation to the ascending orbit interferogram (Figure 6a); from exterior to the interior, in ascending orbits, the sequence of color is yellow–blue–red, and in descending orbit, the color sequence is red–blue–yellow. There were asymmetric distribution characteristics of the deformation field of the Jiuzhaigou Ms 7.0 earthquake. The deformation characteristics of the ascending and descending data showed inconsistency: the deformation of the ascending orbit (Figure 6c) showed that the deformation of 0.10 m towards from the satellite occurred in the northwest regions of the epicenter, and deformation with a maximum value of 0.20 m away from the satellite occurred in the southeast area of the epicenter. In the deformation results of the descending orbit data (Figure 6d), the deformation with a maximum value of 0.14 m towards the satellite occurred in the northwest area of the epicenter, and the deformation with a maximum value of 0.19 m away from the satellite occurred in the southeast area of the epicenter. The opposite deformation generated from InSAR image of two orbits indicates that the surface deformation caused by the earthquake is dominated by horizontal deformation, which is in agreement with the main characteristics of strike-slip seismic deformation.

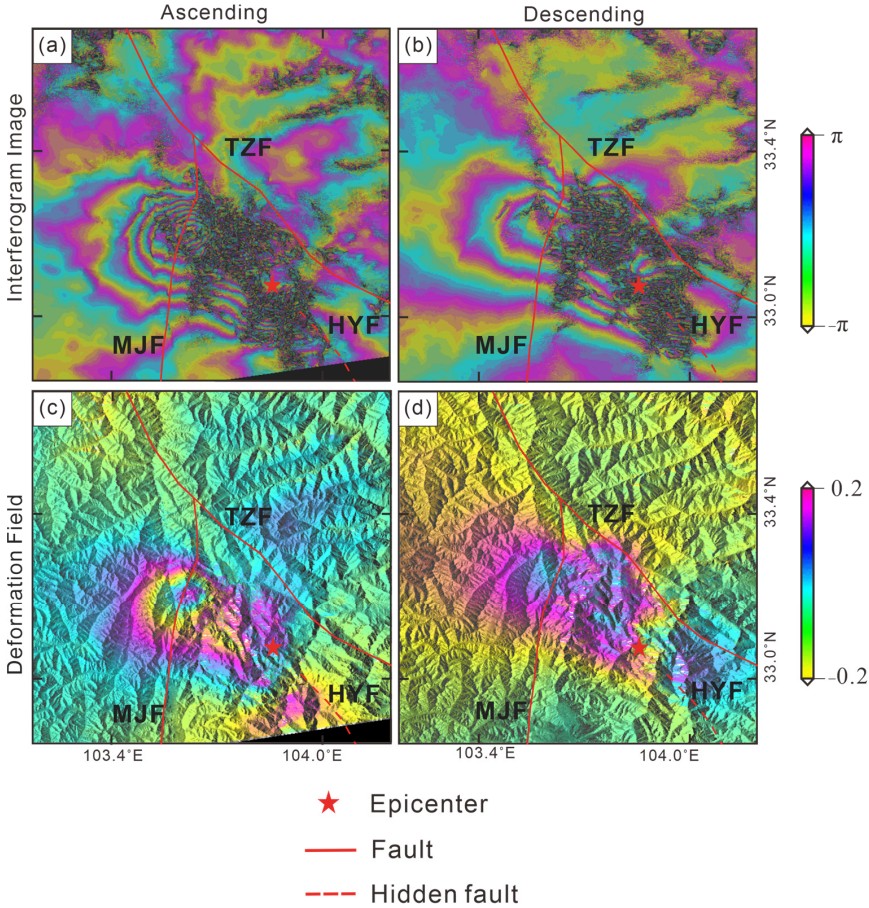

**Figure 6.** Interferogram image and deformation field of the Jiuzhaigou earthquake. Panels (**a**,**b**) are interferogram images of InSAR along ascending and descending orbits, respectively; (**c**,**d**) represent the deformation fields of InSAR along ascending and descending orbits, respectively.

*4.2. SPF Processed Results*

In the gray image (Figure 7a,b), which is identical to the colored image (Figure 6a,b) but in black and white, we can clearly see that the phase increases from the exterior to the interior [22]. The boundary between one ambiguity and the next is shown by the sudden transition from a maximum value (in white) to a minimum value (in black). According to Figure 7, in the original image (Figure 7a,b), the interference stripes are blocked by noise that may be due to excessive vegetation or atmospheric phase. This could make it difficult to judge the distance and direction and azimuth between the interference stripes. After SPF processing, compared with the original interferogram (Figure 7a,b), the boundary lines between the interference fringes (Figure 7c,d) are enhanced. The relationship between adjacent fringes is more obvious. The density of the fringes can be clearly observed. In addition, the SPF method significantly reduces the noise in the interference fringe area (dense black dot area), improves the clarity of the interference stripes, and reduces the incoherence noise caused by a spatial or temporal baseline. The continuity of the interference circle is improved. This could be helpful for further observation and analysis. In the image after the SPF processing, it can be clearly seen that the number of stripes on the west side of the epicenter is higher than that on the east side, indicating that the west side has a larger deformation gradient. Although different imaging modes will inevitably cause the interference fringes and deformation to show different spatial characteristics and magnitudes, it can still be seen from Figure 7c,d that the interference fringes are all butterfly-shaped with obvious four-quadrant patterns, which is a typical strike-slip earthquake deformation characteristic [23].

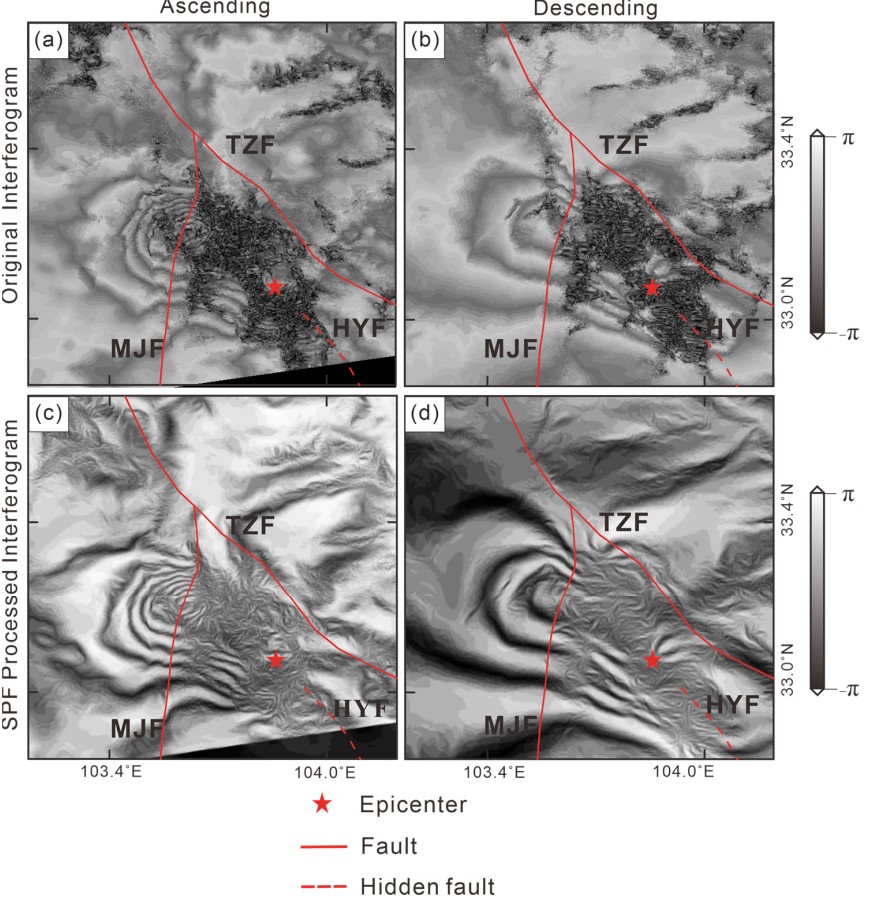

**Figure 7.** Original interferogram and SPF processing on the interferogram. Panels (**a**,**b**) are original interferogram image in grayscale of InSAR along ascending and descending orbits, respectively; (**c**,**d**) represent interferogram images after SPF processed along ascending and descending orbits, respectively.

The interferogram enhanced by the SPF method was unwrapped and geocoded, and the obtained deformation field is shown in Figure 8.

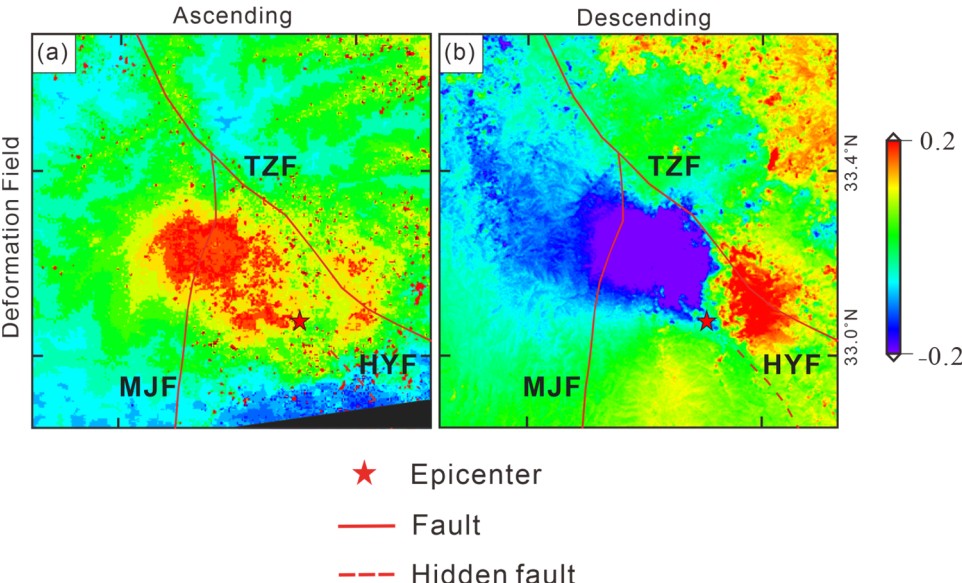

**Figure 8.** Displacement unwrapped from the interferogram image enhanced by SPF method. Panel (**a**) is the deformation field unwrapped from the interferogram in ascending orbit; (**b**) is the deformation field unwrapped from the interferogram in descending orbit.

After unwrapped from the interferogram image enhanced by SPF method, the clarity of the deformation field image is significantly improved, and the boundary of the deformation field becomes clearly visible, making it easier to identify. The maximum deformation area of the descending coseismic deformation field is basically in agreement with that in the ascending deformation field. Since the geometric features of the descending-orbit observation images are different from those of the ascending-orbit observation images, the deformation of the ascending orbit (closer to the satellite) and the deformation of the descending orbit (away from the satellite) may be more than simple uplift and subsidence motions. According to the deformation, the values obtained from the deformation field unwrapped from the SPF enhanced interferogram image (Figure 9a,b) are basically consistent with those obtained from the deformation field unwrapped from the original interferogram (Figure 9c,d). According to the deformation field unwrapped from the interferogram after SPF processing, in the LOS direction, for ascending orbit, the maximum deformation closer to the satellite is 0.11 m, and the maximum deformation away from satellite is 0.20 m; the deformation field is approximately elliptical (Figure 9a). For descending orbit (Figure 9b), in the LOS direction, the deformation of surface towards the satellite is 0.19 m, and the maximum deformation away from the satellite is 0.14 m. The coseismic deformation field is bounded by the Tazang fault zone on the northeast side, between the Tazang fault zone and the Minjiang fault zone, and develops along the end of the Huya fault zone [24]. From the preliminary analysis of the InSAR coseismic deformation situation and the relative spatial distribution of existing faults, the Minjiang fault has no cutting or disturbing effect on the deformation field across it, and should not be a seismogenic fault. Whether the Tazang fault or the Huya fault are seismogenic faults needs to be further determined according to the time series InSAR result, also verifying the effectiveness of the SPF method.

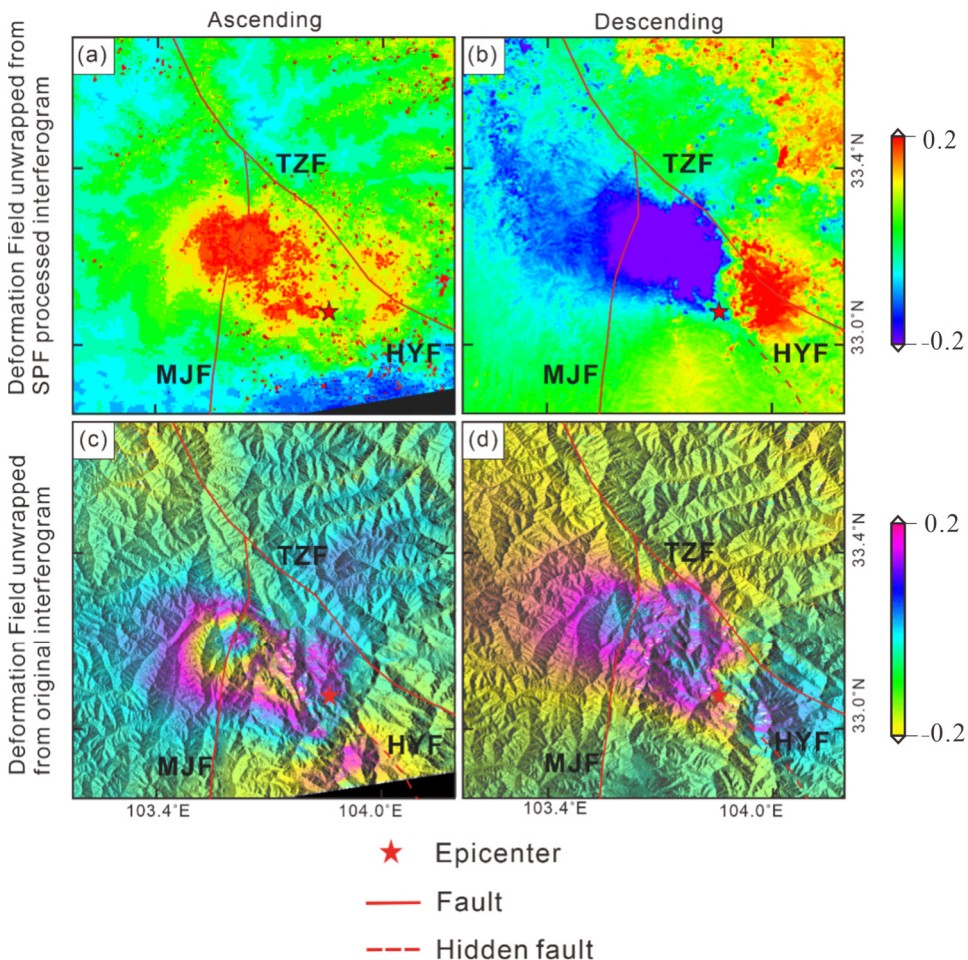

**Figure 9.** The comparison between the deformation field unwrapped from original interferogram and from SPF enhanced interferogram. Panels (**a**,**b**) represent the deformation fields unwrapped from the SPF enhanced interferogram along ascending and descending orbits, respectively; (**c**,**d**) represent the deformation fields unwrapped from original interferogram along ascending and descending orbits, respectively.

According to Table 3, to verify the effectiveness of the SPF method, this paper evaluates the deformation unwrapped from the original interferogram and that unwrapped from the SPF enhanced interferogram results by using the remaining residual points, the unwrapping running time, edge preservation index (EPI) and the root mean square error (RMSE).

**Table 3.** The quantitative evaluation of the results before and after using the SPF method.

|  | SPF Method | Original |
| --- | --- | --- |
| Number of residual points remain | 194 | 227 |
| Unwrapping time/s | 1.02 | 4.70 |
| EPI | 0.65 | 0.37 |
| RMSE | 0.242 | 1.235 |

The quantitative evaluation of the results of the SPF enhanced and original interferogram is shown in Table 4; the number of residual points of the SPF processed interferogram is smaller, and the unwrapping time is shorter, which means less computation, and the RMSE of the deformation unwrapped from the SPF processed interferogram is closer to 0. The EPI of the SPF processed images is large, which means that the edge preservation ability of SPF method is better.

**Table 4.** Results of line-of-sight deformation of Jiuzhaigou earthquake by different researchers.

| Organization | Satellite | Orbit | Uplift/m | Subsidence/m |
|---|---|---|---|---|
| Nie et al. | Sentinel-1A | Ascending | 0.07 | 0.21 |
| | | Descending | 0.16 | 0.08 |
| Ji et al. | Sentinel-1A | Ascending | 0.11 | 0.22 |
| | | Descending | 0.09 | 0.10 |
| Chen et al. | Sentinel-1A | Ascending | 0.07 | 0.21 |
| | | Descending | 0.16 | 0.08 |
| Shan et al. | Sentinel-1A | Ascending | 0.10 | 0.22 |
| | | Descending | 0.14 | 0.10 |
| This essay | Sentinel-1A | Ascending | 0.10 | 0.20 |
| | | Descending | 0.19 | 0.14 |

### 4.3. Validation Analysis of the Deformation Field

The epicenter of the Jiuzhaigou earthquake is located in a mountainous area, and conventional observation techniques based on ground observation station data are difficult to implement. The distribution of observation stations near the epicenter is sparse; there are only GNSS J416 and JB33 stations and GPS 51JZZ and 51JZB stations in the study area, and the number of observation points is small. Many scholars have obtained the LOS deformation data of the Jiuzhaigou coseismic deformation field [23–26], and the inversion results are shown in Table 4. Due to the differences in satellite types, operation direction, spatiotemporal baseline, the data processing methods, etc., there are definite differences in the deformation results. According to Table 4, the results of the deformation intervals in line of sight obtained from ascending orbits are the following: the deformation towards the satellite is 0.07~0.11 m, and the deformation away from the satellite is 0.18~0.22 m. The results obtained from the descending data are the following: the interval of the deformation towards the satellite is 0.10~0.16 m, and the interval of the deformation away from the satellite is 0.05~0.10 m. Compared with the results of different organizations, the results of this paper, whether from original data or the data after SPF processing, are all within the interval, and the deformation results obtained can be considered credible.

### 4.4. 3D Displacement Decomposition Constraints from MAI Method

Since the D-InSAR method is fundamentally a side-looking radar measurement technique, the result is not the true deformation of the surface, but the projection of the true deformation vector of the surface in the line-of-sight direction through a series of coordinate system transformations and geometric transformations [27]. Retrieving true surface deformation information requires decomposition of the deformation in line of sight into north–south deformation and vertical deformation. The deformation that does not decompose into the N–S direction is due to the small sensitivity along that direction, which is alignment with the orbit inclination of satellite [11].

After D-InSAR processing, MAI technology is required to extract the deformation of the SAR image pair in azimuth direction (close to the north–south direction), which is important because the displacement of D-InSAR LOS is not sensitive to the north–south displacement component. The MAI method divides conventional one-scene SAR images into forward- and backward-looking SAR images bounded by a zero Doppler center. The two SAR images can be decomposed into two forward-looking images and two backward-looking images, then the four forward-looking and backward-looking images interfere to obtain the interferogram images, and finally the forward- and backward-looking interferograms are conjugated and multiplied to obtain the azimuth interferogram, and azimuth deformation can be obtained. The line-of-sight and azimuth deformation field obtained using the MAI method is shown in Figure 10.

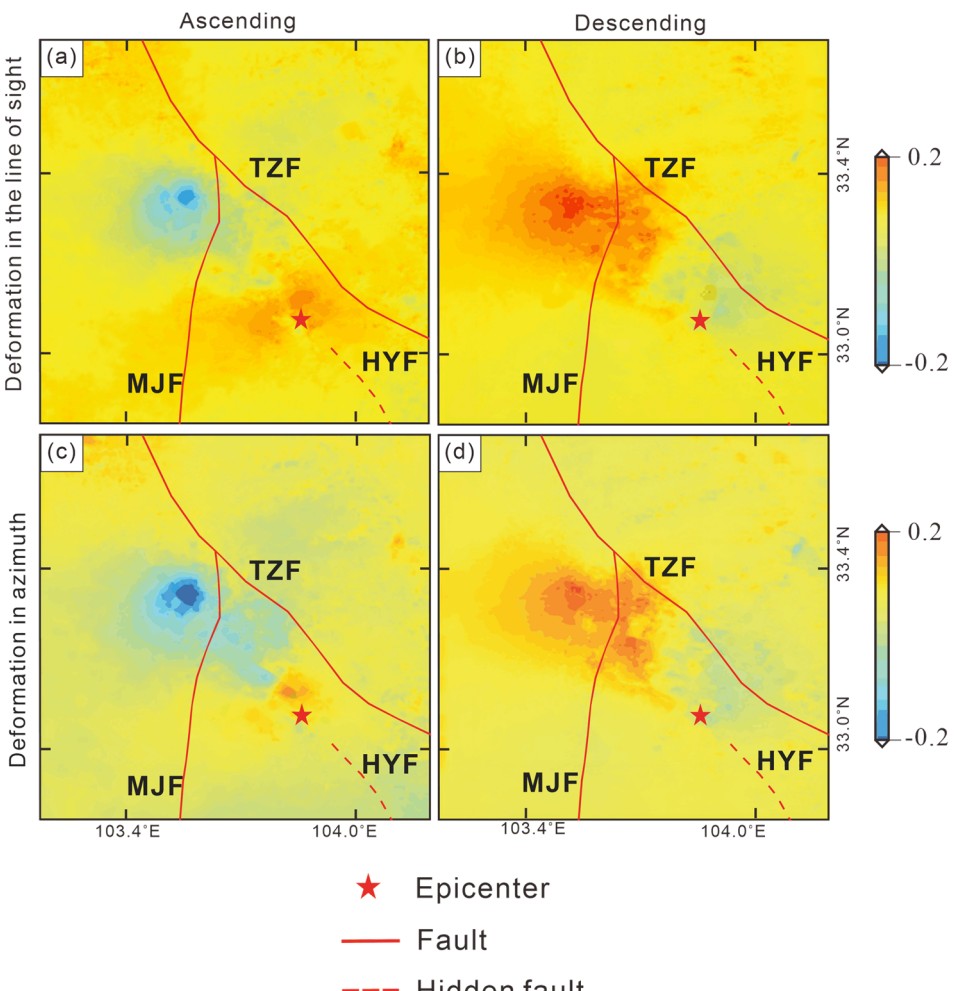

**Figure 10.** Deformation field of the Jiuzhaigou earthquake generated from combined D-InSAR and MAI method. Panels (**a**,**b**) are deformation of InSAR line of sight along ascending and descending orbits, respectively; (**c**,**d**) represent the deformation fields of InSAR azimuth direction along ascending and descending orbits, respectively.

The deformation characteristics of the ascending and descending data show inconsistencies. In ascending orbit, the maximum value of deformation away from the satellite is 0.18 m along the line of sight, occurring in the northwest of the epicenter, and the maximum value of 0.20 m along the azimuth direction occurring in the northwest of the epicenter. The maximum value of deformation towards the satellite along the line of sight was 0.12 m, and that along the azimuth was 0.1 m in the southeast of the epicenter. In the descending orbit, a maximum deformation towards the satellite occurred in the northwest area of the epicenter with a value of 0.14 m along the line of sight and a maximum value of 0.14 m along the azimuth direction. The maximum deformation away from the satellite is 0.11 m along the LOS and 0.13 m along the azimuth in the southeast area of the epicenter.

To decompose the deformation field, the ascending orbit incidence angle $\theta_A$, the descending orbit incidence angle $\theta_D$, the ascending orbit azimuth angle $\alpha_A$, and the descending orbit azimuth angle $\alpha_D$ are brought into the coefficient matrix $C$ of the observation equation:

$$C = \begin{bmatrix} cos\theta_A & sin\alpha_A sin\theta_D & -cos\alpha_A sin\theta_A \\ cos\theta_D & sin\alpha_D sin\theta_A & -cos\alpha_D sin\theta_D \\ 0 & cos\alpha_A & sin\alpha_A \\ 0 & cos\alpha_D & sin\alpha_D \end{bmatrix} \tag{10}$$

Then, we have the coefficient matrix $Z$ of the three-dimensional deformation field $X$:

$$Z = \begin{bmatrix} 0.646 & 0.646 & 0.093 & -0.094 \\ -0.0001 & 0.0001 & 0.513 & -0.513 \\ -0.717 & 0.717 & -0.259 & -0.258 \end{bmatrix} \tag{11}$$

Thus, the three-dimensional coseismic deformation components of the surface are obtained:

$$d_V = 0.636 I_{losA} + 0.646 I_{losD} + 0.093 I_{azA} - 0.093 I_{azD} \tag{12}$$

$$d_N = -0.0001 I_{losA} + 0.0001 I_{losD} + 0.513 I_{azA} - 0.513 I_{azD} \tag{13}$$

$$d_E = -0.717 I_{losA} + 0.717 I_{losD} - 0.259 I_{azA} - 0.258 I_{azD} \tag{14}$$

The decomposed three-dimensional coseismic deformation field is shown in Figure 11.

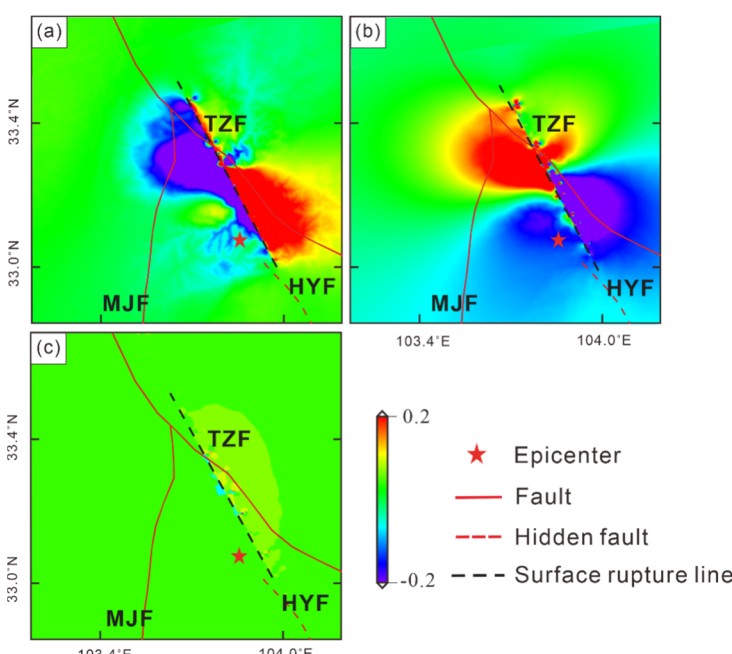

**Figure 11.** Three-dimensional coseismic deformation field of Jiuzhaigou earthquake. Panel (**a**) is the vertical component of the deformation field, (**b**) is the east–west component of the deformation field, and (**c**) is the north–south component of the decomposition of the deformation field. The black dotted line is the position of the surface rupture in this earthquake.

After decomposition, it can be seen from Figure 11 that the coseismic deformation in the north–south direction (Figure 11c) was small, the vertical deformation (Figure 11a) contributions near the epicenter were relatively average, and there were deformations of 0.07 m uplift on the southeast side of the epicenter and 0.10 m subsidence on the northwest side. The coseismic deformation variable in the east–west direction (Figure 11a) is large, and there is a clear displacement division, with a maximum deformation of 0.23 m to the east on the northwest side of the source center and 0.13 m to the west on the southeast side. The deformation in the northwest side (downthrow) of the fault rupture is large, which mainly slides to the southeast; the deformation in the southeast side (upthrow) is smaller than that of the northwest side.

After obtaining the line-of-sight and azimuth deformation information, this paper will inverse-simulate the LOS deformation information of the Jiuzhaigou earthquake based on the LOS deformation, the source parameter interval published by GTM, and the Okada theory.

### 4.5. Fault Slip Distribution Inversion and Analysis

The Okada [28] model is used to construct the functional relationship between ground deformation data and underground fault parameters, so as to simulate the observed interference deformation field and estimate the fault parameters. The main steps of inverting the coseismic slip distribution of faults include downsampling processing of deformation field data, nonlinear inversion of fault geometric parameters (longitude, latitude, strike, dip, slip angle, depth, fault length, width, and slip), and linear inversion of fine slip distribution on fault planes [29]. The initial fault geometric model (strike about 150° and dip angle of 40° to 70°) was constructed, models with different dip angles were tested, and a single plane fault model with strike and dip angles of 150° and 50° was finally determined; the fault area was 45 km × 40 km. In order to better obtain the slip distribution details of the fault plane, this paper divides the fault plane into several subfaults of 2 km × 2 km [30]. Before inversion, the InSAR deformation field is masked by the coherence coefficient, obtaining high-quality deformation data points with coherence greater than 0.3, and then it uniformly samples these data points.

The inversion results of the coseismic slip distribution show that the fault movement is generally left-lateral strike-slip, and the main structural line of the overall shape is controlled by the NW-oriented nodal plane. The inversion results (Figures 12 and 13) show that the earthquake rupture is mainly concentrated in the depth range of 1~15 km underground, the maximum slip of the earthquake is 0.77 m, it occurs at ~9 km underground, the seismic moment is 3.98E+18 N* m, and the moment magnitude is Ms 6.6.

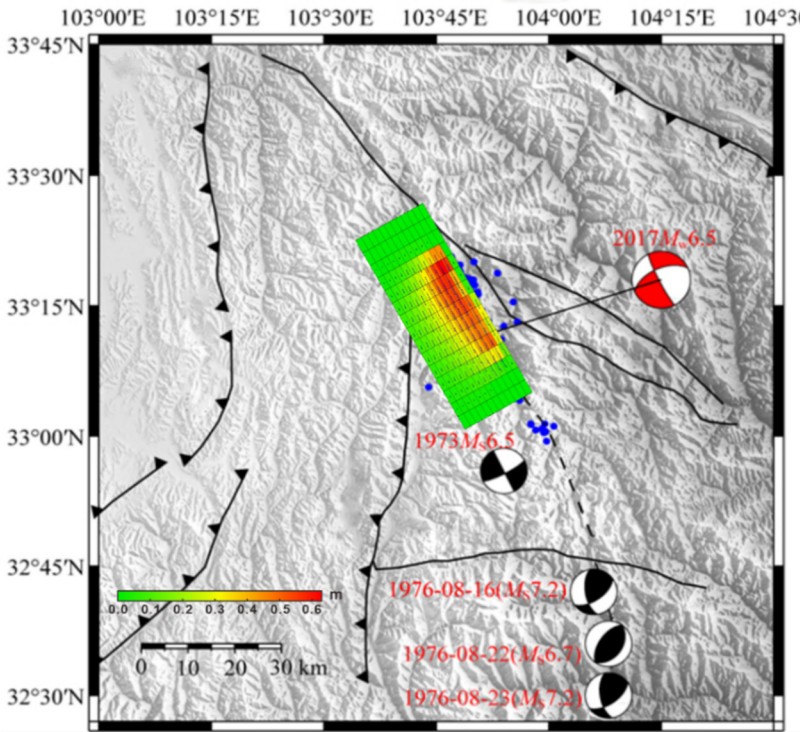

**Figure 12.** Fault slip distribution from inversions constrained by InSAR.

It shows that the fault rupture movement is dominated by left-lateral strike-slip, the focal depth is about 9 km, the strike is 266.49°, the dip angle is 41°, and the slip angle is −160.74°. The comparison between the inversion results of Harvard University, United States Geological Survey (USGS), and the China Earthquake Network Center (CENC) is shown in Table 5. The epicenter location is basically the same as that of other institutions, the magnitude is basically the same as that of other institutions, the focal depth is slightly smaller than that of other institutions, and the strike and slip angle are similar to those of

other institutions. The institutions are basically consistent, which proves the reliability of the results of this paper from this side.

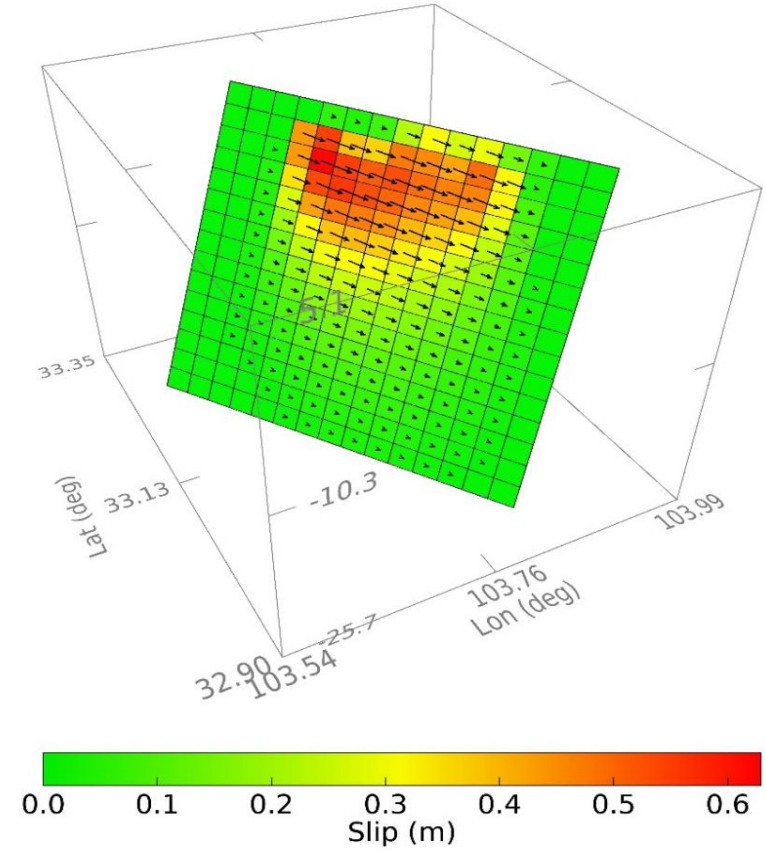

**Figure 13.** Fault slip distribution from InSAR inversions shown in 3D.

**Table 5.** Earthquake mechanism from different earthquake institutions.

| Institution | Longitude/°E | Latitude/°N | Magnitude | Depth/km | Strike/° | Slip/° |
|---|---|---|---|---|---|---|
| Harvard | 103.9 | 33.21 | 6.5 | 14.9 | 242 | −168 |
| CENC | 103.82 | 33.20 | 6.5 | 11 | 326 | −15 |
| USGS | 103.85 | 33.193 | 6.5 | 9 | 246 | −173 |
| This Paper | 103.81 | 33.16 | 6.6 | 9 | 267 | −160.74 |

In order to validate the credibility of the results, the inversion fault model is used to fit the coseismic InSAR deformation field, and the fitting results and residual distribution are shown in Figure 14. The correlation between the observation data and the model reached 90.8%, and the main deformation characteristics were well fitted. There are certain residual errors in the near-field region of the fault model, which may be related to factors such as atmospheric error, DEM error, simplification of the fault model, and unwrapping error. From the residuals' distribution, it can be seen that most of the residuals are less than 8 cm. The results show that the simulated deformation field is basically in agreement with the observed deformation field in terms of distribution shape and magnitude, which shows the reliability and rationality of the inversion in this paper.

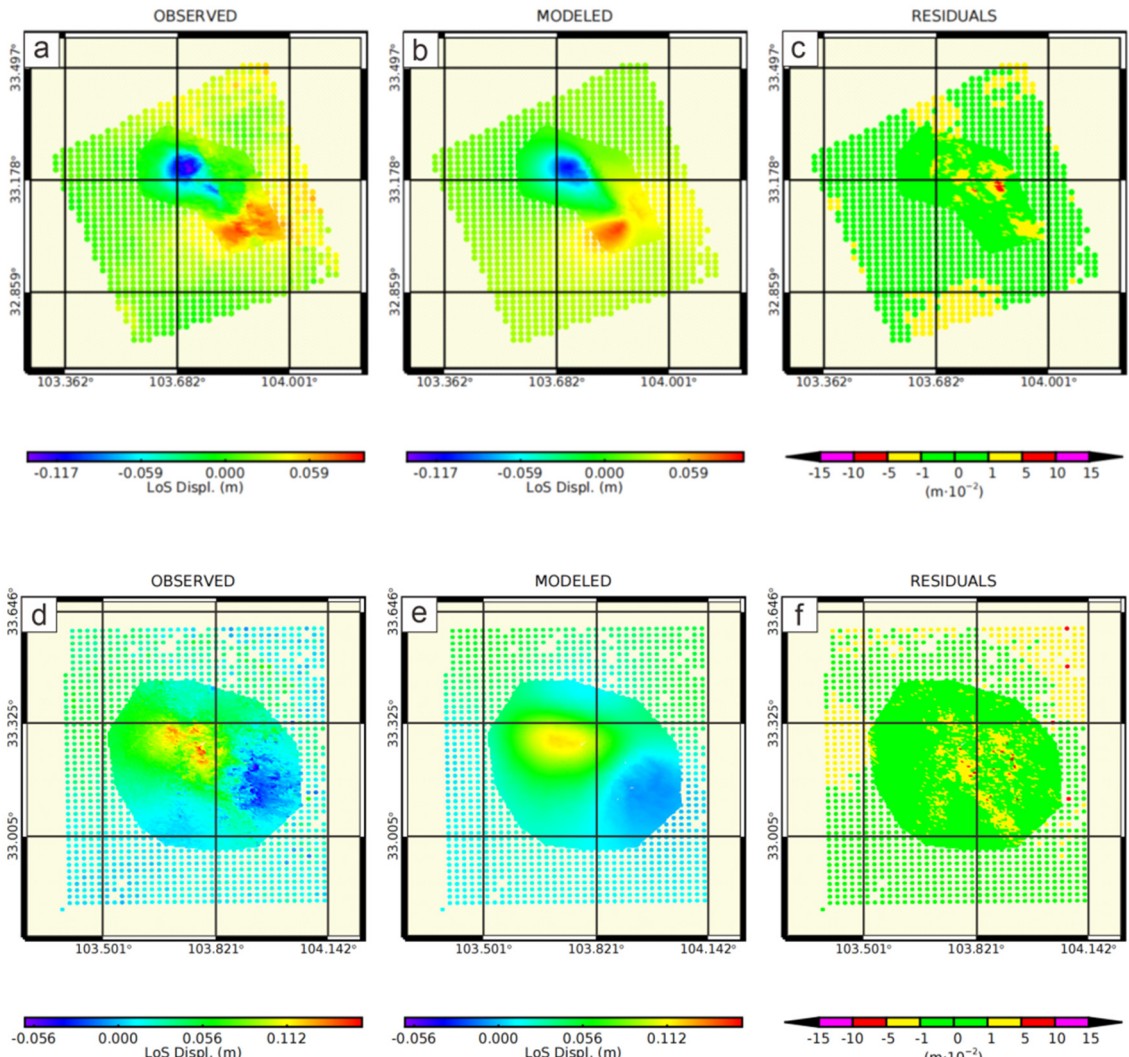

**Figure 14.** Observed, modeled deformation field and residuals inverted from InSAR data: Panels (**a**,**b**) are observed and modeled ascending InSAR deformation fields; (**c**) is the residuals to the observed deformation field; (**d**,**e**) are observed and modeled descending InSAR deformation fields; (**f**) is the residuals to the observed deformation field.

### 4.6. SBAS Time Series Displacement Velocity Map

Figure 15 shows the overall vertical average velocity in this area obtained by SBAS technology in this study. The southern part of the study area has a larger deformation range, but the deformation rate is relatively stable, and the annual deformation rate reaches −50–20 mm/a. In addition, there are some smaller areas in the central and northern regions, but the subsidence rate is relatively fast; their area is less than 1 square kilometers, but the fastest subsidence rate exceeds 70 mm/a. In order to further explore the change of the overall deformation of the study area with time, a total of 10 sampling points in three areas were selected for analysis (Figure 15) [31]. The information of the sampling area and the distribution of sampling points are shown in Table 6.

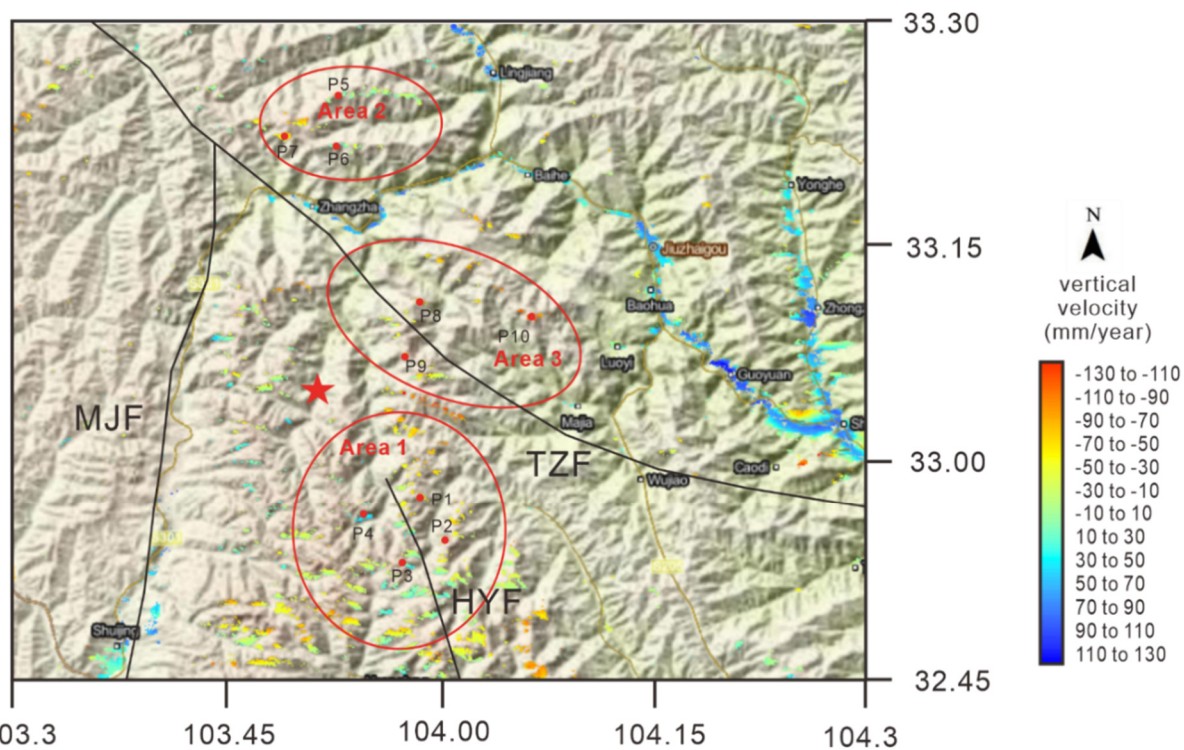

**Figure 15.** Jiuzhaigou earthquake 20170818–20180226 average vertical velocity map.

**Table 6.** Sampling area information and sampling point distribution information.

| Area | Central Coordinate | Point | Longitude | Latitude |
|---|---|---|---|---|
| | | 1 | 103°97″ | 32°99″ |
| 1 | 103.96E 32.94N | 2 | 104°03″ | 32°89″ |
| | | 3 | 103°96″ | 32°90″ |
| | | 4 | 103°92″ | 32°96″ |
| | | 5 | 103°89″ | 33°42″ |
| 2 | 103.9E 33.39N | 6 | 103°88″ | 33°36″ |
| | | 7 | 103°82″ | 33°37″ |
| | | 8 | 103°97″ | 33°19″ |
| 3 | 103.98E 33.17N | 9 | 103°96″ | 33°14″ |
| | | 10 | 104°01″ | 33°18″ |

Area 1 (Figure 16) occupies most of the study area and is located on the hidden extension line of the Huya fault. In order to explore the law of surface deformation in this area, this paper extracts four sampling points: p1 and p2 are located in the hanging wall of the Huya fault, and p3 and p4 are located in the footwall of the fault. Among them, p1 and p4 are closer to the epicenter, and the disturbance after the earthquake is more obvious. It is shown in Figure 16 that there are still large fluctuations after the earthquake, before December 2017. The difference between the maximum uplift and the maximum subsidence value of p4 is 24.8 mm, and that of p1 is 27.4. After December 2017, the deformation rates of p1 and p4 gradually stabilized. Combining the subsidence rate curve and the average rate map, it can be seen that the points marked by p1 and p2 on the upper wall of the Huya fault are displayed in red and yellow, showing the subsidence in the line of sight (satellite observation direction), while the lower wall contains the blue of p3 and p4. The characteristics of the deformation field obtained separately by the ascending and

descending orbits are opposite, which is usually a sign of post-earthquake deformation caused by strike-slip faults, so the Huya fault is likely to be a seismogenic fault [25].

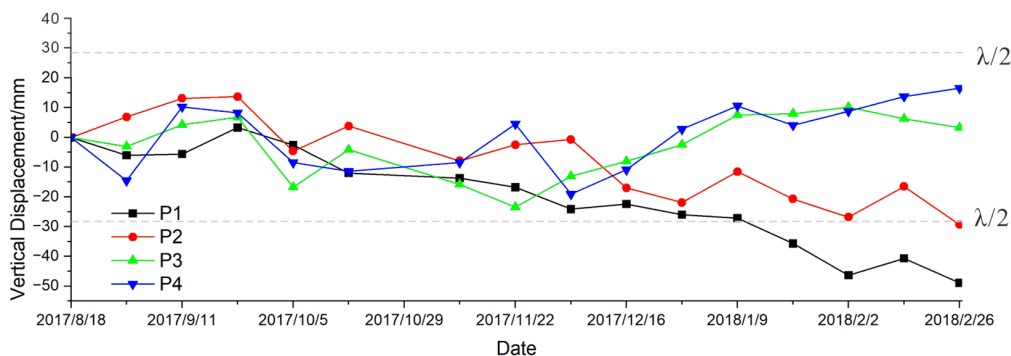

**Figure 16.** Time series deformation diagram of sampling points in region 1. The gray dotted lines are ambiguity (wavelength/2).

Area 2 (Figure 17) is located in the northeastern part of the study area and is located north of the intersection of the Tazang and Minjiang faults. This area shows overall subsidence in Figure 17, and the deformation rate is mostly between −30 and 10 mm/a. In this area, three points (P5–P7) were selected as sampling points for time series analysis. It can be seen from Figure 18 that from 2017-08-18 to 2017-09-23, the area was in a state of uplift due to the disturbance of the earthquake. The accumulated subsidence of the three points was observed from 2017-09-23 to 2017-12-16 and amounted to 28.87 mm, 28.92 mm and 32 mm, respectively. After 2017-12-16, the region began to stabilize, and its deformation fluctuation was less than ±5 mm.

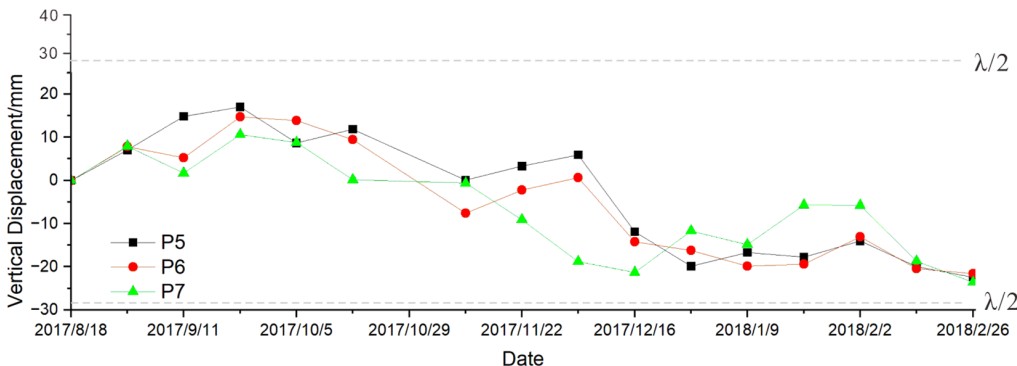

**Figure 17.** Time series deformation diagram of sampling points in region 2. The gray dotted lines are ambiguity (wavelength/2).

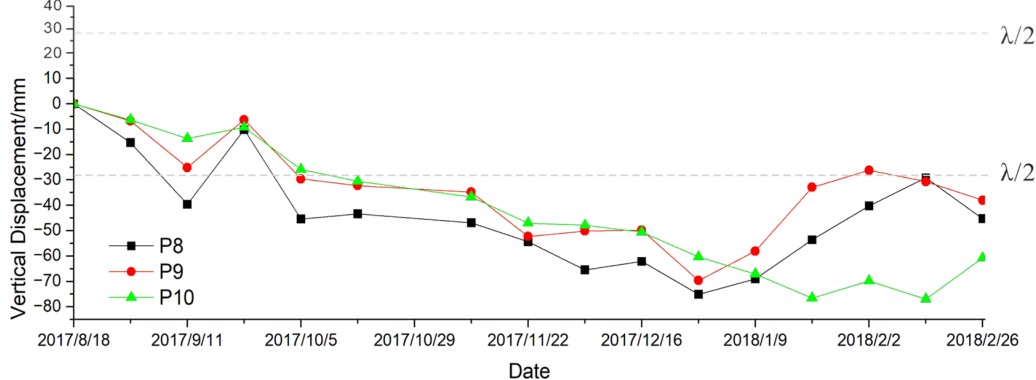

**Figure 18.** Time series deformation diagram of sampling points in region 3. The gray dotted lines are ambiguity (wavelength/2).

Area 3 (Figure 18) is located above the Tazang fault, and is at the north of the extension line of the Huya fault. It can be seen from Figure 18 that from 2017-08-18 to 2017-12-28, this area is in a state of rapid subsidence, and its accumulated subsidences, respectively, reach 75 mm, 69.6 mm and 60.3 mm. All points show the characteristics of subsidence, and it can basically be judged that the subsidence occurred at the place affected by the earthquake. No obvious deformation and relative displacement of the upper and lower plates of the Tazang fault were observed before and after the earthquake, which is evidence that the Tazang fault is not a seismogenic fault for the Jiuzhaigou earthquake [32].

### 4.7. Analysis of the Spatiotemporal Movement Characteristics of the Huya Fault Segment

Based on the LOS time series deformation field, this study analyzes the coseismic surface motion characteristics of the Jiuzhaigou earthquake area, which is helpful to study the motion characteristics of related faults and provide a reference for subsequent earthquake risk assessment [1]. In order to understand the detailed movement deformation rate of the Huya fault and its temporal and spatial variation characteristics across the fault, based on the LOS-trending time series deformation field, eight representative section lines (from A–A′ to H–H′) were selected along the fault strike direction (Figure 19), and the LOS-direction deformation rate profile of the SBAS point at each dividing line was drawn to analyze its motion characteristics.

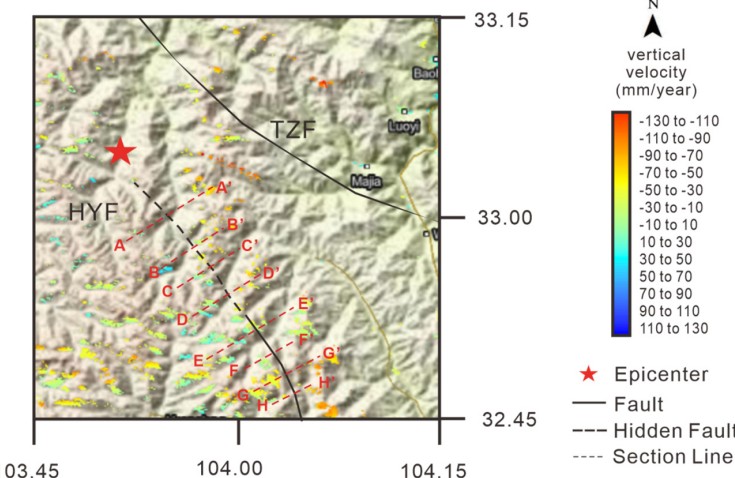

**Figure 19.** The annual average deformation rate field of the Huya fault.

In this study, the crustal deformation and fault movement characteristics of the eastern Himalayan tectonic junction based on time series InSAR for each cross-fault profile line of the Huya fault are plotted as a graph, as shown in Figure 20. The horizontal axis in the figure represents the distance between the point and the fault line (the lower left corner of the section line is the starting point), the vertical axis represents the deformation value of each point, blue straight lines represent the intersection of each section line and the fault line, and the dark blue dotted lines are fitted curves of each point shape variable.

The post-earthquake deformation rate of points on A–A′ and B–B′ sections are relatively stable, between −20~20 mm/y with a small subsidence center at the intersection of the fault and section lines, and the relative slip rate of the north and south walls is almost zero. The post-earthquake subsidence rate of each point on profile B–B′ has no obvious trend of change. The points on profile C–C′ show obvious differences in subsidence rates between the two sides of the fault, the south side shows a gradual decrease in the subsidence near the fault, and the north side shows a gradual increase in the subsidence near the fault; the relative slip rate of the north–south walls of the fault reaches 9.93 mm/y. The point displacement rate pattern on the D–D′ profile is just opposite to that on C–C′, the south wall increases while the north wall decreases, and the relative slip rate of the north–south walls reaches 15.65 mm/y. The overall deformation rate of the E–E′ profile

fluctuates between −20 and −60, and the difference between the deformation rates of the north and south plates near the fault is 25.1 mm/y. The deformation rate of the F–F′ profile generally increases from south to north, between −40 and 10. The G–G′ profile has an abrupt subsidence point near the fault, which is thought to be an outlier caused by the deformation of water vapor close to the river [33]. Aside from the outliers, the deformation of this profile increases gradually along the profile line. The deformation rate of each point along the H–H′ section is characterized by subsidence–uplift–subsidence, and a subsidence minimum value of about −40 mm/y appears near the fault.

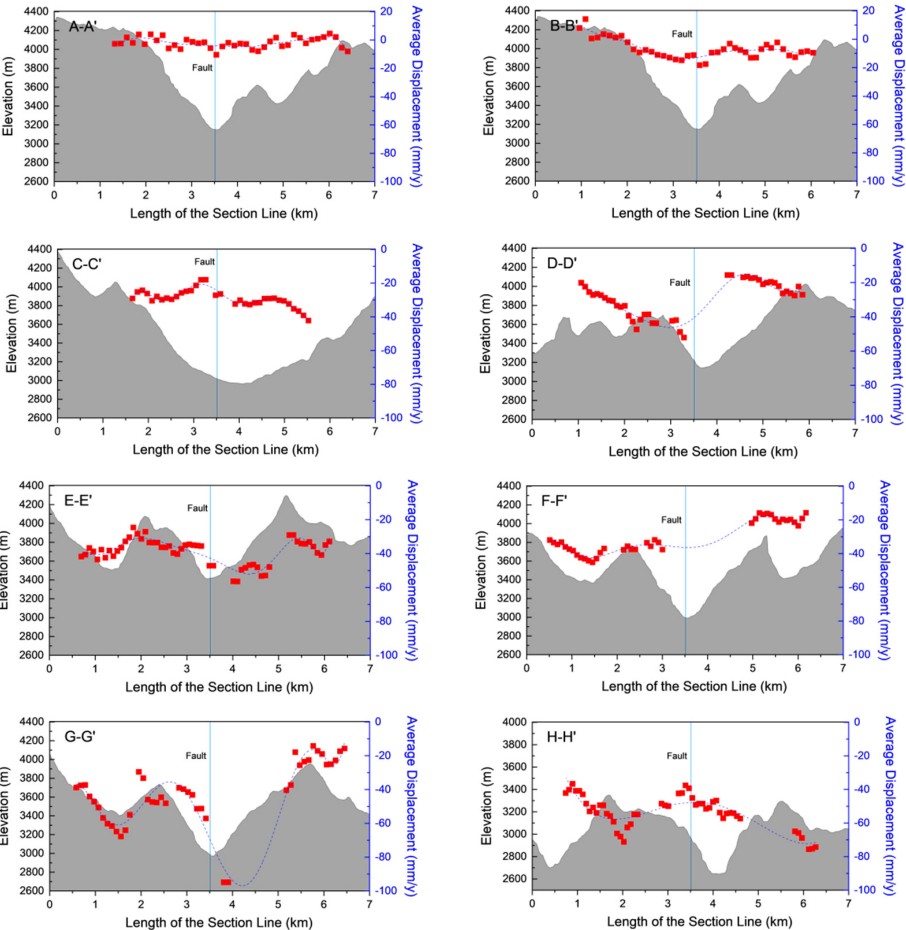

**Figure 20.** Velocity profile of 8 sections of the Huya fault along the fault strike.

Comprehensive analysis of the deformation rate characteristics of the Huya fault section line AA′–HH′ shows that the relative slip rates of the north and south sides increased significantly, and there was an obvious uplift center near the fault. From the fault strike to the middle section, the average deformation rate is about −32.7 mm/a, indicating that the subsidence rate of the fault gradually increases, the north and south sides of the fault move relative to each other, and the north side is uplifted relative to the south side. The deformation rate of the northwestern segment of the fault is within –20~20 mm/y. The fault movement in this area is characterized by no relative movement between the two sides of the fault and low deformation rate.

The occurrence and movement properties of the Huya fault have obvious segmentation characteristics. The northern segment strikes NNW, dips to the NE, and has a dip angle of 80°. The movement in the northern section is dominated by left-lateral strike-slip, which is in agreement with the focal mechanism of the Jiuzhaigou earthquake. InSAR observations and field investigations show that the earthquake-generating faults of the Jiuzhaigou earthquake did not rupture to the surface [34]. From the analysis results of deformation rate, the closer to the northwest fault zone, the lower the surface deformation

rate of the Huya fault InSAR, and the smaller the deformation difference across the two walls of the fault.

## 5. Discussion about the Regional Tectonic Movement

The focal mechanism of the Jiuzhaigou Ms 7.0 earthquake demonstrates that the focal depth is about 6.8 km, the strike is 266.49°, the dip angle is 41°, and the slip angle is −160.74°. The dip angle based on InSAR data inversion reflects the local information at the centroid depth, and optimizing the fault structure can improve the reliability of slip inversion results. It can be basically determined that the vertical and strike-slip characteristics of the earthquake-generating fault in Jiuzhaigou are in agreement with the properties of the northern section of the Huya fault, which is the result of the north–west extension of the Huya fault.

The model aims to explain the movement mechanism of the eastern end of the Kunlun fault (Figure 21). From previous studies, it can be seen that the causes of the Jiuzhaigou earthquake are roughly divided into three stages:

- Stage 1: The left-lateral slip along the main Kunlun fault was transferred to the West Tazang fault through the Huahu basin, which is a pull-apart basin [35].
- Stage 2: A small part of this left-lateral slip is transformed into a southwest-trending reverse movement. The majority of the slip is transformed into the shortening of the crust along nearly-N–S-trending structures, such as the Longriba fault, the Minjiang fault, and the Huya fault. The shortened crust led to the uplift of the Minshan Mountains [36].
- Stage 3: The results of deep seismic reflection show that the Tazang fault, the Minjiang fault, and the Huya fault, pushed by the boundary thrusting [37], are connected with the detachment body at a depth of about 30 km. The connection eventually led to the Jiuzhaigou earthquake.

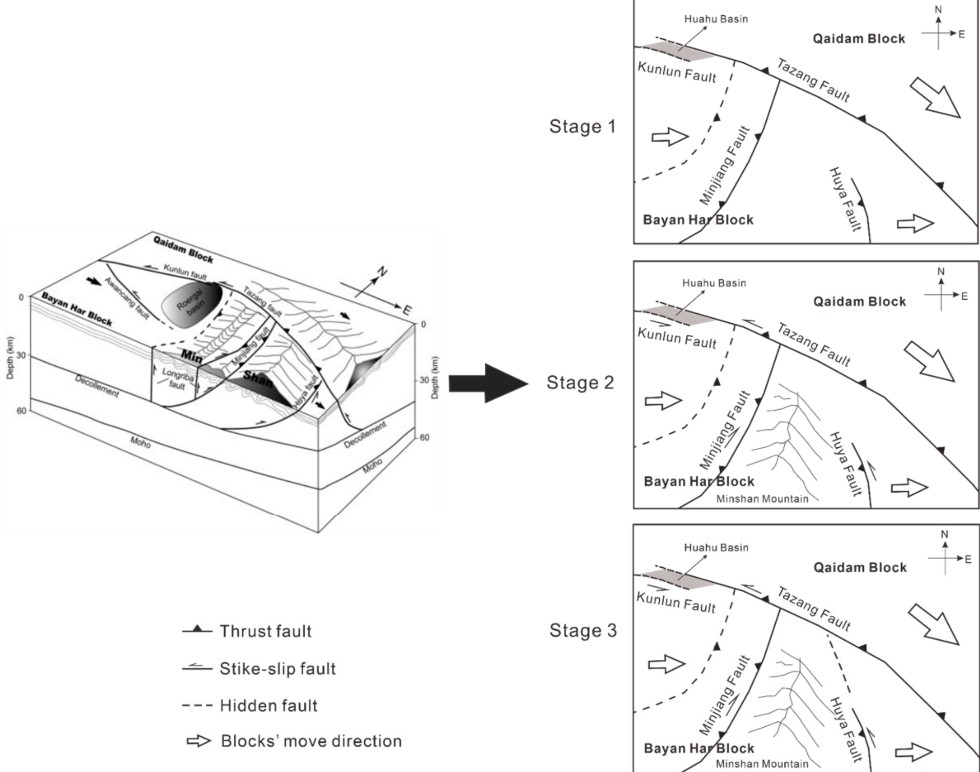

**Figure 21.** Block diagram of the eastern termination of the Kunlun fault. Modified from Ren et al., 2013b [34]. The crustal structure is from deep seismic reflection results. Black arrows show the direction of block motion relative to Eurasia.

Due to its thrust-dominated nature, the left-lateral strike-slip was transformed into the east–west extrusion of the Minjiang and Huya faults, resulting in the Minjiang and Huya faults since the Holocene. Such intense tectonic activity has made strong earthquakes and small earthquakes active from ancient times to the present.

The 2017 Ms 7.0 Jiuzhaigou earthquake occurred on a hidden fault on the eastern boundary of the Bayan Har block, and the overall movement was left-lateral strike-slip, which was in agreement with the left-lateral strike-slip movement of the fault zone at the northern boundary of the Bayan Har block. It again shows the strong stress–strain accumulation state and seismicity characteristics in the eastern boundary area of the Bayan Har block.

## 6. Conclusions

Based on the Sentinel-1 SAR image data provided by ESA, this paper used D-InSAR technology to obtain the synthetic deformation field of the Jiuzhaigou earthquake, used the SPF method to enhance the filtering image, and used this as a constraint to invert the synthetic sliding distribution of the earthquake, Finally, we obtained the cumulative deformation of the study area within one year after the earthquake by SBAS-InSAR technology, and discussed the seismic structure based on the plate motion characteristics. The study found the following:

1.  The steerable pyramid filtering method can be used to analyze the effectiveness of filtering images at different scales, remove invalid components, and enhance the contrast of InSAR images and the continuity of interference fringes. It not only effectively suppresses the noise in the interferogram, but also enhances the edge detail information in the interferogram.
2.  The InSAR interferogram after SPF method processing clearly observed the coseismic deformation field of the Jiuzhaigou earthquake, the maximum amount of surface deformation caused by the Jiuzhaigou earthquake was about 20 cm (line of sight), and there were asymmetric distribution characteristics of the homogenic deformation of the ascending orbit. The 2017 Jiuzhaigou earthquake was dominated by left-lateral slip, the surface movement was dominated by horizontal deformation, the vertical deformation was small, and the coseismic deformation variable in the east–west direction was the largest, with a maximum deformation of 0.2 m to the east and 0.14 m to the west. The maximum sliding amount was about 77 cm, located at a depth of 9 km. The moment magnitude obtained by inversion was Mw 6.6.
3.  The Jiuzhaigou earthquake occurred in the area where the Bayankara block was strongly deformed by the block of the South China block during its south–east movement, and its seismic fault was the hidden part of the north–west extension of the Huya fault.

Currently, the application of the SPF method in D-InSAR interferogram and deformation fields is a new research direction for the combination of geodetic measurement and geophysics. This article intends to provide a new InSAR image processing method, and provide more accurate details for interpreting remote sensing images. The results of SPF applied to D-InSAR images are basically in agreement with the previous research results of geodetic inversion, which further proves the effectiveness of the SPF method in the application of InSAR images, and provides a new long-distance, noncontact method for future fault research.

**Author Contributions:** Conceptualization, W.P. and X.H.; methodology, X.H.; software, Z.W.; validation, W.P., X.H. and Z.W.; formal analysis, W.P.; investigation, W.P.; resources, W.P.; data curation, W.P.; writing—original draft preparation, X.H.; writing—review and editing, W.P.; visualization, X.H.; supervision. All authors have read and agreed to the published version of the manuscript.

**Funding:** This research received no external funding.

**Data Availability Statement:** Not applicable.

**Acknowledgments:** The SAR images acquired by Sentinel-1A were downloaded from the Copernicus Open Access Hub and the NASA Distributed Active Archive Center at the Alaska Satellite Facility (https://earthdata.nasa.gov/eosdis/daacs/asf) (accessed on 10 September 2021). The Shuttle Radar Topography Mission (SRTM) DEM with a resolution of about 90 m/pixel were downloaded from the National Aeronautics and Space Administration (https://data.nasa.gov) (accessed on 29 March 2022).

**Conflicts of Interest:** The authors declare no conflict of interest.

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
