# Peer review of "Coseismic Deformation and Fault Inversion of the 2017 Jiuzhaigou Ms 7.0 Earthquake: Constraints from Steerable Pyramid and InSAR Observations"

_remotesensing, doi:10.3390/rs15010222_

Round 1
Reviewer 1 Report
This is a well-written research paper with clear and understandable language, clear figures, and tables that represent accurately the results and the study design is appropriate for answering the research questions. Both Summary and Conclusions are clearly written, giving accurate information on the research and the results, without spin for the reviewer and the reader as long as it gets published and gets accessed to a broader audience.
The issues I‘ve found reading the paper are of minor importance. On pages 10 (under Figure 9), 11, and 12 the word co-seismic is highlighted in yellow,
Overall, the soundness of the methodology and the conclusions can be supported by the results, with a relatively novel approach in seismic ambient-noise imaging which has a significant impact on the field.
Therefore, I recommend this research paper to be published after some minor changes would be carried out.
Kind regards
Author Response
Dear reviewer:
We appreciate very much for your positive comments and suggestions on our manuscript, here's our response:
Point 1: On pages 10 (under Figure 9), 11, and 12 the word co-seismic is highlighted in yellow.
Response 1: We have canceled the highlighting of words.
Kind regards
Reviewer 2 Report
The paper obtained the synthetic deformation field of the Jiuzhaigou earthquake with D-InSAR technology enhanced by the Steerable Pyramid Filtering method. SBAS InSAR method was used to obtain the cumulative deformation across the fault system and the seismic structure based on the plate motion characteristics were discussed. Here are some weakness which should be improved or strengthened in the manuscript:
(1) In the abstract (from Line 10-11), it is stated that “… were used to derive the surface displacement observations along the satellite line-of-sight (LOS) and azimuth directions using the differential interferometric SAR (InSAR, DInSAR) method”. I wonder how the traditional DInSAR method can measure the azimuthal deformation?
(2) In Line 72-73, “… using The local fringe frequency of the InSAR …”, “The” in the middle of a sentence should be changed to “the”.
(3) In Line 83, the sentence “The purpose of interferometric phase pattern filtering.” is not complete.
(4) The introduction section should be re-organized. A summary of previous work should be replace in the second and third paragrahs in this section (Line 47-94). The work conducted in this paper should put together and be clarified (Line95-107 and Line117-121).
(5) It is suggested that the title of Section 2.2 changes to “data and processing”, because in this section, both data and processing are addressed.
(6) Please check the track number in Table 1 in Page 5, “T55” or “T128”. The table 1 is duplicate with Line 275-287.
(7) Section 3.1 and 3.2 did not clarify the basic principles of the DInSAR and SBAS methods. They are more like introductions of these two methods which focus on their history and characteristics.
(8) In Line 224, the expression and spelling of PS-InSAR (“that is, the points with better coherence in PS-SAR”) were not precise.
(9) Line 275-287 were duplicated with Section 2.2. I don’t know why?
(10) Figure 9 and Figure 10 should be improved. The same color bar should be used so that the comparison would be more clear. Besides, both 0.2 and -0.2 were shown in the same color in Fig. 10 which will confuse the readers.
(11) In Line 296, Fig. 9a should be Figs. 9a and 9b.
(12) In Line 297, Fig. 9b should be Figs. 9c and 9d.
(13) In Figure 13, it seems that the inversion of the coseismic slip distribution is based on the ascending deformation field. What about the inversion result of descending pair? Are they consistent?
(14) In Line 403, as mentioned above, the seismogenic fault is generally a left-lateral strike-slip fault, so the interseismic deformation may still along the horizontal direction. Why in Line 403, the author interpreted the descending LOS displacement as vertical deformation? I would suggest using “near/far-range” to signify the surface motion.
(15) In Line 411, “Table 4” in the end of the paragraph should be “Table 5”?
Author Response
Dear reviewer:
We appreciate very much for your positive and constructive comments and suggestions on our manuscript, the response is uploaded as the attachment.
Kind regards

Reviewer 3 Report
Dear authors, thank you for your manuscript. I regards the assessment of earthquake movements using InSAR, using steerable filtering method to enhance the (noisy) interferograms.
I am an InSAR specialist and I cannot judge the discussion regarding the earthquake effects and tectonic movement.
Language: sections 3.1 and 3.2 need to be re-written due to general incomprehensibility. However, unsuitable expressions can be found in the whole manuscript, so I think a language review would be useful.
Section 3.3: I am missing the explanation of "theta direction" and what does the "f" represent.
line 342: you are referring to "small black circle" in fig. 10c, but I could not find any
Do I understand well that you process the ascending data and descending data, but do not perform the decomposition into vertical and east-west component? Do you expect the movement to have also the north-south component?
Therefore, the time series in figures 15-17, 19 are in satellite line of sight?
Regarding the "outliers" in figure 19 G-G', could this be (partially) caused by the fact that it is in line of sight?
For the time series in figures 15-17, I suggest to plot the ambiguity (wavelength/2), so that possible unwrapping errors are visible. It may also explain some steep post-earthquake changes.
Author Response

(The authors gave the same response as above.)

Reviewer 4 Report
The manuscript addresses an interesting topic and is adequately developed, both with the adequate exposition of the methodological procedures and the technological resources used. The results obtained are interesting for several purposes, but need adequate validation. Thus, 2 points stand out and deserve to be added to the manuscript: first, in terms of the validation (Were the data obtained validated in the field or with data from instruments installed in the region? What types? What is the spatial distribution?) and second, if there are records in engineering works that help in the validation of the data obtained. These are not clear in the manuscript!

Author Response

(The authors gave the same response as above.)

Round 2
Reviewer 2 Report
1. In the revised manuscript, the author added the 3D displacement decomposition from LOS measurements (Section 4.4 and Fig. 9), but there are no detailed descriptions about the decomposition process. Since there are only two observations (ascending and descending), it is impossible to decompose them into three directions without any presupposition. I wonder how the author can get the 3D deformation? It seems that the author calculated the 3D displacement by dividing the LOS measurement (from one observation) by the cosine of the angle in each direction, which is clearly wrong. On the premise of neglecting the north-south deformation, it is feasible to invert the 2D displacement (vertical and east-west) from the combination of ascending and descending LOS measurements. But in this study, the north-south component seems too large to ignore. It is recommended to study the following literature to understand the decomposition process.
Fuhrmann, T., & M. C. Garthwaite (2019). Resolving three-dimensional surface motion with InSAR: Constraints from multi-geometry data fusion. Remote Sensing, 11(3), 241. https://doi.org/10.3390/rs11030241
2. The author used the SPF method to enhance the interferogram (Fig. 7) and displacement field (Fig. 8), which seem to be implemented separately, but I wonder if the two results are credible and consistent. Maybe the author can try to get the displacement field by unwrapping the SPF-enhanced interferogram, and compare the result with the SPF-enhanced displacement field (Fig. 8).
Author Response
Dear Reviewer,
Thanks for your comments, and we have checked the manuscript throughout and also revised the manuscript according to your suggestion. The response is in the attachment.
We look forward to hearing from you regarding our submission. We would be glad to respond to any further questions and comments that you may have.
Kind regards
Wenshu Peng

Reviewer 3 Report
Dear authors,
thank you for the revised manuscript.
Most of my comments were taken into account. I am just sending comments for the rest of them, from which only the first is important.

Author Response

(The authors gave the same response as above.)
